# Chemical Solution Deposition of Barium Titanate Thin Films with Ethylene Glycol as Solvent for Barium Acetate

**DOI:** 10.3390/molecules27123753

**Published:** 2022-06-10

**Authors:** Sabi William Konsago, Katarina Žiberna, Brigita Kmet, Andreja Benčan, Hana Uršič, Barbara Malič

**Affiliations:** 1Electronic Ceramics Department, Jožef Stefan Institute, Jamova Cesta 39, 1000 Ljubljana, Slovenia; katarina.ziberna@ijs.si (K.Ž.); brigita.kmet@ijs.si (B.K.); andreja.bencan@ijs.si (A.B.); hana.ursic@ijs.si (H.U.); 2Jožef Stefan International Postgraduate School, Jamova Cesta 39, 1000 Ljubljana, Slovenia

**Keywords:** BaTiO_3_ thin films, solvent influence, chemical solution deposition, crystallinity, ferroelectric, piezoelectric

## Abstract

Chemical solution deposition (CSD) of BaTiO_3_ (BT) or BT-based thin films relies on using a carboxylic acid and alcohol as the solvents for alkaline-earth carboxylate and transition-metal alkoxide, respectively; however, the esterification reaction of the solvents may lead to in-situ water formation and precipitation. To avoid such an uncontrolled reaction, we developed a route in which ethylene glycol (EG) is used as the solvent for Ba-acetate. The EG-based BT coating solutions are stable for at least a few months. The thermal decomposition of the BT xerogel obtained by drying the EG-based solutions depends on the choice of the solvent for the Ti-alkoxide as well: in the case of EG and 2-methoxyethanol solvents carbon residues are removed at only about 1100 °C, while in the case of ethanol it is concluded at about 700 °C. About 100 nm thick BT films derived from the EG-ethanol solution deposited on platinized silicon reveal dense, crack-free columnar microstructure. They exhibit local ferro- and piezoelectric properties. The macroscopic polarization-electric field loops were obtained up to a quite high electric field of about 2.4 MV/cm. The EG-ethanol based CSD route is a viable alternative to the established acetic acid–alcohol route for BT and BT-based films.

## 1. Introduction

Among lead-free piezoelectric materials, barium titanate-based materials, especially Ba(Zr_0.2_Ti_0.8_)O_3_-(Ba_0.7_Ca_0.3_)TiO_3_ solid solution or BZT-BCT, show outstanding piezoelectric properties with a *d_33_* coefficient exceeding the value of Pb(Zr,Ti)O_3_ (PZT) based materials [1,2,3]. Materials in the form of thin film are considered for the miniaturization of devices. In the BZT-BCT formulation, barium titanate (BT) is the main component, so we can consider it as the reference material for studies on the BZT-BCT. On the other hand, BT is the end-member of (Ba, Sr)TiO_3_ solid solution that has been proven to be a good candidate for tunable microwave devices as well [4,5,6].

One of the most commonly used methods to fabricate BT-based thin films is chemical solution deposition (CSD). This method offers several advantages, such as low capital cost, control of the stoichiometry of the coating solutions, simple and fast preparation procedures, the possibility of large-scale deposition, and control of the thickness of the coatings [6,7,8,9]. We note that in the majority of CSD routes organic solvents have been used. Recently, environmentally-friendlier water-based synthetic approaches have been reported, which mainly focused on deposition on single-crystal perovskite substrates [10,11].

In CSD of BT as well as BT-based thin films, alkaline earth carboxylates and transition metal alkoxides are commonly used as starting materials to prepare the coating solutions [12,13,14,15]. Normally, alkaline-earth metal carboxylates are dissolved in carboxylic acids, and alkoxides are diluted with an alcohol [6,8,16]. As a side reaction, the slow interaction between alcohol and carboxylic acid, i.e., the esterification reaction leads to the progressive in-situ formation of water (Figure 1).

Transition metal alkoxides are very sensitive to water, leading to hydrolysis (Figure 2) and, hence, the precipitation of hydroxides.

In the case of BZT-BCT, a higher reactivity of zirconium alkoxides towards water compared to titanium may increase the possibility of precipitation [17,18]. Indeed, our preliminary experiments on CSD of BZT-BCT revealed that the temporal stability of conventionally synthesized solutions was only a few weeks. A possible solution would be to reduce the reactivity of zirconium alkoxide by adding chelating agents such as acetylacetone, 3-hydroxy-2-butanone or certain amine compounds [16,19]. However, the esterification reaction and thus the possibility of hydrolysis would not be impeded.

Another possibility would be to replace acetic acid with another solvent for the alkaline earth reagent to avoid the esterification reaction that could lead to the formation of water in the coating solutions. Ethylene glycol (EG) was used to dissolve lead acetate in CSD of PZT films [20]. We found that EG can dissolve Ba-acetate as well, which has not been reported earlier. Otherwise, EG also serves as a solvent and chelating agent for Ti alkoxide [21,22].

In this work, we established a systematic study on CSD of BT films using EG as the solvent for barium acetate instead of acetic acid. We note that in this case, the solvent for the transition metal alkoxide was 2-methoxyethanol as reported in [6]. BT served as the reference material for a more complicated BZT-BCT solid solution. Furthermore, we also introduced different solvents for the transition-metal alkoxide. In addition to 2-methoxyethanol, we used EG as the common solvent for both metal compounds and ethanol. Furthermore, we optimized the heat treatment conditions.

We find that the BT solutions containing EG, the alkaline-earth acetate solvent, are stable for months. The thermal decomposition of the dried solution that contains ethanol is concluded at a lower temperature than the solutions containing EG or 2-methoxyethanol as the solvent for the alkoxide. The films deposited from the EG-ethanol based solution yield perovskite films with columnar microstructure and good dielectric and ferroelectric properties.

## 2. Results and Discussion

### 2.1. Thermal Decomposition of BT Xerogels

The BT coating solutions prepared with different solvents were dried at 200 °C for 12 h to yield xerogels. The gels and thin films obtained from respective solutions, namely acetic acid-2-methoxyethanol, ethylene glycol, ethylene glycol-ethanol and ethylene glycol-2-methoxyethanol are denoted as AcOH-MOE, EG, EG-EtOH and EG-MOE.

The FTIR spectra of the gels collected in Figure 1 reveal the absorption bands at 1565 cm^−1^ and 1419 cm^−1^ corresponding to acetate groups in agreement with [7,23]. The FTIR spectra of the EG, EG-EtOH and EG-MOE gels contain weak bands at about 3500 and 2900 cm^−1^, which we attribute to the presence of trace EG residues. The boiling point of EG is 197 °C and it is possible that the solvent or its residues were not fully evaporated at 200 °C from the gel network which is not the case for the AcOH-MOE specimen.

The thermal decomposition of the xerogels was analyzed by thermogravimetry and differential thermal analysis (TG and DTA) coupled with evolved gas analysis (EGA). TG, DTA and EGA curves of the AcOH-MOE, EG-EtOH, EG and EG-MOE gels are shown in Figure 2 and Figure 3. In all samples, a small mass loss upon heating to about 100 °C accompanied by a slight endotherm and the evolution of water evident in the EGA curves is attributed to evaporation of adsorbed humidity. Likewise, in EGA, trace amounts of acetone (CH_3_COCH_3_) were recorded in all samples upon heating around 400 °C, see Appendix A. Acetone is a side product in the decomposition pathway of the acetate groups as documented in the literature [24,25,26].

The thermal decomposition of the AcOH-MOE sample takes place upon heating to 730 °C and is characterized by a two-step weight loss of 32.68%. The first weight loss of 19.01% from 280 °C to 402 °C is accompanied by an exothermic peak and simultaneous evolution of H_2_O and CO_2_ which indicates the thermal oxidation of organic groups [26]. In the interval from 620 °C to 734 °C, the weight loss of 13.67% is accompanied by a series of weak exothermic peaks and the evolution of CO_2_ corresponding to carbonate decomposition. The decomposition pathway of the AcOH-MOE sample agrees with earlier reports [23].

The thermal decomposition of the EG-EtOH gel is concluded upon heating to 719 °C. The weight loss of 18.49% from 280 °C to 422 °C is marked by a series of exothermic peaks, with the strongest one at 402 °C and the simultaneous evolution of H_2_O and CO_2_. We observe a slight weight loss upon heating to about 580 °C where the evolution of both H_2_O and CO_2_ indicates the decomposition of traces of organic residues. Between 585 and 719 °C, an exothermic weight loss of 13.02% is due to carbonate groups decomposition. Clearly, the decomposition pathways of the AcOH-MOE and EG-EtOH samples show similarities, the organic groups are decomposed in the first step. The second step of mass loss in these two samples is accompanied by a relatively small exothermic peak in DTA and by the evolution of CO_2_ due to the decomposition of carbonate that is concluded at about 700 °C.

In contrast, the thermal decomposition processes of the EG and EG-MOE samples collected in Figure 3 are concluded at much higher temperatures, 1076 °C, and 1127 °C, respectively. Upon heating from room temperature, we observe progressive weight loss with a series of exothermic peaks and evolution of H_2_O and CO_2_ peaks up to about 600 °C. Between 600 and 800 °C, only CO_2_ is evolved indicating carbonate groups decomposition.

Upon further heating to the final temperature of 1200 °C, there is another weight loss, upon which CO_2_ is evolved.

It is worth mentioning that the important difference between these two samples and the AcOH-MOE and EG-EtOH is that in the former group the final mass loss is concluded only at about 1100 °C. We expect that the oxide material prepared from AcOH-MOE or EG-EtOH could be consolidated at lower temperatures than that prepared from EG or EG-MOE solvents as carbonaceous residues are decomposed at about 400 °C lower temperature.

To support the thermal analysis data, the xerogels were calcined at 900 °C for 15 min. The FTIR analysis revealed the presence of metal-oxygen bonds in the AcOH-MOE and EG-EtOH samples. The obvious band at wavenumber 1415 cm^−1^ in the spectra of EG-MOE and EG samples is attributed to the presence of carbonate groups [7], see Figure 4a. Quite similarly, Ashiri detected weak carbonate bands in IR spectra of BT gels heated up to the final temperature of 800 °C [23]. For the EG-MOE and EG samples, a higher calcination temperature was required to yield the oxide phase only, as evident from Figure 4b.

### 2.2. Phase Composition, Microstructure and Properties of BT Thin Films

We spin-coated four BT solutions on platinized silicon substrates, dried and pyrolyzed them at 250 °C for 2 min, and at 350 °C for 2 min, and annealed them at 800 °C. The coating-heat treatment procedure was repeated four times. Multiple annealing steps were introduced to enable crystallization of individually deposited layers with the thickness of a few 10 nm via heterogeneous nucleation and, consequently, the formation of predominantly columnar microstructure in agreement with earlier studies [27,28].

Figure 5 shows the XRD patterns of BT thin films. All films crystallize in the perovskite phase (JCPDS 96-154-2141). We note that the peak of the (111) crystallographic plane coincides with the (111) Pt peak of the substrate. In all films, the (110) peak is the most intense, especially in the AcOH-MOE and EG-MOE; however, it should be noted that the Pt (111) peak coincides with the BT (111) peak. A comparison of crystallite sizes calculated from the (110) peak broadening using the Scherrer equation (see Appendix A) reveals almost two times larger values compared to the crystallite sizes of the EG-EtOH and EG films, about 90 nm versus 55–60 nm.

In Figure 6a–d, the plan-view and inset the cross-section micrographs of AcOH-MOE, EG, EG-MOE, and EG-EtOH BT films are collected. The average thickness of all films is about 100 nm, and the cross-section images reveal predominantly columnar microstructures. The plan-view images reveal dense microstructures with grains in the range of 50–100 nm across. However, in the AcOH-MOE and EG-MOE films, obvious intergranular cracks are visible on the surface, which is not the case for the EG and EG-EtOH films. It is worth mentioning that these four films were prepared under the same conditions, so the cracks are presumably not associated with the material formulation, film thickness or annealing, or phase-transition temperature-related thermal stresses. We speculate that the cracks are due to the larger grain and crystallite sizes of the AcOH-MOE and EG-MOE films (see Appendix A). In (Ba,Sr)TiO_3_ thin films, the appearance of intergranular cracks was related to the combined effects of increased grain size and films thickness [6]. Local electrical measurements show that all these films are locally ferroelectric/piezoelectric active (see PFM images and local hysteresis loops in Appendix A). The dielectric permittivity and losses of the EG-EtOH film measured at 1 kHz and room temperature were 495 and 0.2, respectively, while the measurements of other films could not be reliably performed which we mainly attribute to the presence of intergranular cracks.

In the process of selecting the optimum EG-based coating solution formulation we considered two arguments:-The final temperature of the thermal decomposition of the xerogels was in the case of EG and EG-MOE gels at about 1100 °C to ensure complete removal of carbonaceous residues, while for the EG-EtOH gel the thermal decomposition was concluded at a much lower temperature, about 700 °C, cf. Figure 2 and Figure 3.-Intergranular cracks developed in the EG-MOE film while the EG and EG-EtOH films were crack-free.

In further work, we thus focused on the EG-EtOH films and compared them with the AcOH-MOE films which served as reference.

Additionally, we increased the time of drying and pyrolysis steps of the EG-EtOH and AcOH-MOE films from 2 min to 15 min so that we would ensure the complete removal of residual organics. Namely, we observed that we needed quite long drying times of coating solutions (12 h) to obtain reproducible thermal decompositions of xerogels, cf. Figure 2 and Figure 3. Furthermore, earlier reports on CSD of BT films indicated quite long times for pyrolysis, for example, 7.5 min at 250 °C [29]. We prepared the films consisting of one deposited layer only, and recorded FTIR patterns after the drying at 250 °C for 15 min, pyrolysis at 350 °C for 15 min, and rapid-thermal annealing steps at 400 °C, 500 °C, 600 °C and 700 °C. The acetate groups evidenced by the absorption bands at about 1430 cm^−1^ and 1580 cm^−1^ for (COO-) symmetric and (COO-) asymmetric stretching vibrations could be identified in the dried and pyrolyzed films, see Figure 7, in agreement with [23]. In the AcOH-MOE film, the band of the carbonate group was present upon heating to 600 °C, and only at 700 °C did it disappear. The FTIR patterns of propionate-derived BT thin films contained carbonate bands upon annealing to 650 °C while at 700 °C only metal-oxygen bands were observed [7]. In the EG-EtOH film, the carbonates were decomposed between 500 and 600 °C, about 100 °C lower than in the AcOH-MOE films.

In Figure 8, XRD patterns of AcOH-MOE and EG-EtOH films prepared with 15 min drying pyrolysis times and annealed at 800 °C, consisting of four deposited layers are collected. The films crystallize in the perovskite phase (JCPDS 96-154-2141). We note that the AcOH-MOE film is strongly (110) oriented. The crystallite size of the AcOH-MOE film is 91 nm, indicating only a slight increase compared to the film with shorter drying/pyrolysis times (87 nm). A significant increase in the crystallite size is observed in the EG-EtOH film if the drying/pyrolysis time increases from 2 to 15 min, namely from 59 nm to 76 nm (see Appendix A).

The plan-view and cross-section microstructures of the AcOH-MOE and EG-EtOH films are collected in Figure 9. In the surface microstructure of the AcOH-MOE film, some intergranular cracks are observed, while the EG-EtOH film is crack-free. The grain size is 50–100 nm in both cases. The cross-sections of about 100 nm thick films consist of predominantly columnar grains. In comparison with the films prepared with the short, 2 min drying and pyrolysis steps, these films seem to have more pronounced columnar microstructures.

A more detailed investigation of the AcOH-MOE and EG-EtOH films microstructure was performed by transmission electron microscopy (TEM). In Figure 10a,b, the cross-section microstructures of AcOH-MOE and EG-EtOH films, respectively, reveal predominantly columnar grains of about 50–100 nm across, in agreement with the SEM analysis. Such microstructure formed as a result of sequential crystallization of individually deposited layers on top of pre-crystallized layers in the process of multistep annealing [27]. In some parts, we also observe smaller equiaxed grains. In Figure 10c,d, the selected area diffraction (SAED) patterns are shown together with the TEM images of the analyzed regions. According to the diffraction data, grains in both films are preferentially [110] and [111] oriented.

Local piezoelectric/ferroelectric properties of AcOH-MOE and EG-EtOH films were investigated by PFM analysis. Figure 11a,b show the topography, PFM amplitude and PFM phase images of 3 × 3 μm^2^ area scans of AcOH-MOE and EG-EtOH thin films. We note that the topography (deflection mode) of both films agrees with the SEM micrographs (Figure 9). From AFM height images it can be seen that AcOH-MOE and EG-EtOH films are flat, with surface roughness values (Rq) of 2.1 nm and 2.0 nm, respectively. The peak-to-valley values determined from the AFM height images shown in Figure 11 are 15.5 nm and 17.9 nm. The PFM amplitude and phase images of both films show different contrast under the applied voltage indicating the local piezoelectric/ferroelectric response. Local PFM amplitude and phase hysteresis loops of AcOH-MOE and EG-EtOH samples show the typical ferroelectric/piezoelectric behavior (see Figure 12a,b) indicating that both films are locally ferroelectric.

The dielectric permittivity of the AcOH-MOE film at 1 kHz and room temperature is 536 while the value for the EG-EtOH film is 604, in both cases a slight decrease with increasing frequency is observed. Both values agree with literature data for BT films with similar grain sizes [29,30,31,32]. The dielectric losses are about 3.5% and 3% at 1 kHz in Figure 13.

The hysteresis loop of the AcOH-MOE film in Figure 14 a indicates leakage, which could probably be explained by the presence of intergranular cracks observed in this film (Figure 9). It is noted, nevertheless, that this film exhibited local ferro- and piezoelectric responses (Figure 12) and also local current measurements performed with CAFM confirmed the absence of leakage currents higher than 1 pA (see Appendix A).

In contrast, the EG-EtOH film exhibits a well-developed polarization-electric field hysteresis loop shown in Figure 14 b confirming the macroscopic ferroelectric behavior of the film. We note that the film survived a quite high electric field of about 2.4 MV/cm. At the field amplitude of 1.4 MV/cm, the values of remnant and saturation polarizations are 8 μC/cm^2^ and 26 μC/cm^2^, respectively, and a coercive field of about 275 kV/cm. The values cannot be compared with earlier works where the maximum electric fields were about an order of magnitude lower, e.g., [31,33].

## 3. Materials and Methods

Barium titanate (BaTiO_3_, BT) thin films were prepared by chemical solution deposition (CSD). Barium acetate (Ba(CH_3_COO)_2_, Ba(OAc)_2_) with purity 99.999%, and titanium butoxide (Ti(OC_4_H_9_)_4_, Ti(OnBu)_4_) with purity 99.61% were used as reagents, both purchased from Alfa Aesar, Karlsruhe, Germany. Four coating solutions were prepared by dissolving these reagents in different combinations of solvents: acetic acid (CH_3_COOH, AcOH, 100%, Alfa Aesar, Karlsruhe, Germany), 2-methoxyethanol (CH_3_OCH_2_CH_2_OH, MOE, 99.3+%, Sigma-Aldrich, St. Louis, USA), ethylene glycol (OHCH_2_CH_2_OH, EG 99.5%, Riedel-de Haën, Seelze, Germany) and absolute ethanol (CH_3_CH_2_OH, EtOH 99.9%, Carlo Erba, Val-de-Reuil, France). The manipulation of chemicals and chemical reactions took place in a dry nitrogen atmosphere.

The coating solutions and samples prepared from these solutions are named according to the solvents used for their preparation. The procedures are described below and schematically summarized in Figure 15. In a typical experiment, 25 mL of the 0.2 M Ba-Ti coating solution was prepared.

The AcOH-MOE solution was prepared using AcOH and MOE as solvents following the procedure described in earlier papers [6,28]. Ba(OAc)_2_ was dissolved in AcOH at 60 °C and cooled to room temperature. Ti(OnBu)_4_ was diluted with MOE and mixed for ≥15 min. Then the two solutions were mixed in the equimolar ratio for 2 h at room temperature and the concentration was adjusted to 0.2 M. The AcOH/MOE volume ratio was 3/2.

The solution denoted EG was prepared using EG as the only solvent. Ba(OAc)_2_ was dissolved in EG at room temperature. The Ti(OnBu)_4_ solution in EG was added to the Ba(OAc)_2_ solution in the equimolar ratio. After mixing at room temperature for 2 h the concentration of the Ba-Ti solution was adjusted to 0.2 M. In the solution denoted EG-EtOH, Ti(OnBu)_4_ was diluted with absolute EtOH at room temperature, mixed with the solution of Ba(OAc)_2_ in EG in equimolar ratio for 2 h to yield the 0.2 M Ba-Ti solution. In the EG-MOE solution, Ti(OnBu)_4_ was diluted with MOE instead of EtOH. In EG/EtOH or EG/MOE the volume ratio of respective solvents was 3/2.

BaTiO_3_ films were deposited from the four solutions on Pt(111)/TiO_2_/SiO_2_/(100)/Si substrates (Pt/Si, purchased from SINTEF, Oslo, Norway) by spin coating at 3000 rpm for 30 s (WS -400B-6NPP/LITE, North Wales, PA, Laurell). The as-deposited wet films were dried at 250 °C for 2 min and pyrolyzed at 350 °C for 2 min on hot plates. For selected AcOH-MOE and EG-EtOH films, the times of drying and pyrolysis steps were prolonged to 15 min. The films were annealed at 800 °C with heating and cooling rates of 13,33 °C/s and 2,6 °C/s, respectively, in a rapid thermal annealing furnace (Mila 5000, Ulvac-Riko, Yokohama, Japan) after each deposition-drying pyrolysis step. The annealing times for the first and the last deposited layers were 15 min, and 5 min for intermediate layers. This procedure was repeated four times to achieve a film thickness of about 100 nm. Please note that the films have been named after the coating solutions they were deposited from.

The thermal decomposition of the xerogels obtained by drying the respective coating solutions at 200 °C for 12 h was followed using a simultaneous thermal analyzer coupled with a mass spectrometer (STA 409, Netzsch, Selb, Germany + ThermoStar, Balzers Instruments, Oerlikon, Switzerland). The samples with a mass of about 33 mg were heated in Pt/Rh crucibles with a heating rate of 10 K/min. Thermogravimetric curves (TG), differential thermal analysis (DTA) and evolved gas analysis (EGA) were recorded up to 1200 °C, in a flowing synthetic air atmosphere.

The BT gels, calcined at 900 °C for 15 min and the powder samples obtained after the thermal analysis were collected for Fourier transform infrared spectroscopy (Perkin Elmer FTIR Spectrum 100, Waltham, Massachusetts USA, 4000–380 cm^−1^). FTIR was also used to follow the chemical composition of the AcOH-MOE and EG-EtOH films. The films consisting of one spin-coated layer were dried at 250 °C for 15 min, pyrolyzed at 350 °C for 15 min and then rapid-thermally annealed at 400 °C, 500 °C, 600 °C and 700 °C for 6 s. The FTIR spectrum was recorded after each drying/pyrolysis/heating step.

XRD analysis of the prepared films was performed using a high-resolution diffractometer (X’Pert PRO, PANalytical, Cu Kα radiation, Almelo, The Netherlands) with 2 theta = 10–39, 40–65 deg, step = 0.016, time per step = 100 s, soller slit = 0.02, mask10. Note that the 2-theta range was selected so that the Pt (111) peak was not recorded. The XRD data were analyzed by X’Pert High Score Plus software for the phase analysis.

Scanning electron microscopy analysis was performed using field emission scanning electron microscopes (FE-SEM) JSM-7600F (JEOL, Tokyo, Japan) with 10 kV accelerating voltage and Verios 4G HP (Thermo Fischer, Waltham, Massachusetts, USA). To avoid charging problems during analysis, the samples were pre-coated with 5 nm of carbon using a Precise Etching and Coating System 628A (Gatan, Pleasanton, California, USA).

A detailed investigation of the thin film microstructure was performed using a scanning-transmission electron microscope JEOL ARM 200 CF (JEOL Ltd., Tokyo, Japan) operated at 200 kV. The cross-section specimens were prepared by a classical sandwich technique—the films were cut with a diamond saw, glued face-to-face and mounted in brass rings with epoxy. The samples were then ground, polished and dimpled. Electron transparency was archived by Ar+ milling using a PIPS 691 ion milling system (Gatan, Peasanton, California, USA). 

Piezo-response force microscopy (PFM) and conductive atomic force microscopy (C-AFM) were performed using Jupiter XR and MFP-3D atomic force microscopes (Asylum Research, Santa Barbara, California, USA) equipped with a PFM and C-AFM modules. For PFM and C-AFM scanning, a conductive tip with a diameter of about 15 nm made of Ti/Ir layer coated on Si (Asyelec-01, Atomic Force F&E GmbH, Abingdon, UK) was used. During PFM scanning in dual ac resonance tracking mode, an electric voltage of 3 V and a frequency of ~350 kHz were applied between a conductive AFM tip and the bottom electrode of the samples. The PFM amplitude and phase hysteresis loops were measured in the off-electric-field switching spectroscopy mode with the pulsed dc step signal and the superimposed ac drive signal, as described in [34]. The waveform parameters were: the sequence of increasing dc step signal was driven at 20 Hz; the frequency of the triangular envelope was 0.2 Hz; a superimposed sinusoidal ac signal with an amplitude of 3 V and a frequency of ~350 kHz was used. Three cycles were measured, the second cycle is shown in Figure 12. CAFM measurements were performed with ORCA mode using dc signal of up to 25 V.

Dielectric permittivity and losses as functions of frequency were measured using an impedance analyzer at room temperature (HP 4284A) with a frequency range of 100 Hz to 100 kHz. Ferroelectric hysteresis loops were measured using an Aixacct TF analyzer 2000 (Systems GmbH, Aachen, Germany) with a sinusoidal signal, a frequency of 1 kHz. Gold electrodes with a diameter of 0.4 mm and 0.2 mm were deposited on the films by magnetron sputtering (5 Pascal). The platinum bottom contact was reached by etching the films with a mixture of HF, HCl and H_2_O in the ratio (2:5:20).

## 4. Conclusions

Since early papers on CSD of BT and BT-based films, the synthesis of the coating solution has involved a carboxylic acid as the solvent for alkaline-earth salt and an alcohol as the solvent for the transition metal alkoxide. Such choice of the solvents unavoidably results in slow but progressive in-situ evolution of water, which contributes to uncontrolled hydrolysis of the alkoxide and reduces the stability of the solution.

By replacing acetic acid (AcOH) with ethylene glycol (EG) as the solvent for Ba-acetate the stability of coating solutions was increased from weeks to months. We found that the choice of the alcohol solvent for Ti-alkoxide influenced the course of the thermal decomposition of the xerogels obtained by drying the respective coating solutions. In the case of EG and 2-methoxyethanol (MOE), which can both act as chelating agents the thermal decomposition of the xerogels was concluded only at about 1100 °C. The xerogel obtained from the EG-ethanol based solution was thermally decomposed into an oxide at about 700 °C, which is comparable to the data for the acetic-acid derived xerogel.

BT thin films were deposited on platinized silicon substrates, dried, pyrolyzed and rapid thermally annealed at 800 °C. The about 100 nm thick films crystallized in the perovskite phase with predominantly columnar microstructures. Intergranular cracks were observed in the BT films deposited from the AcOH-MOE and EG-MOE solutions, which could be tentatively related to larger grain and crystallite sizes (about 90 nm) of respective films compared to EG-ethanol derived films (about 60 nm).

The films prepared using the latter solvent formulation were considered for further study and after optimizing the drying-pyrolysis conditions compared to the acetic-acid derived films. PFM analysis revealed that the films show the local piezoelectric/ferroelectric behavior. The room temperature dielectric permittivity values at 1 kHz of the EG-ethanol and AcOH-MOE films are about 600 and 540, in agreement with the literature data. The EG-EtOH film exhibits polarization-electric field loops up to the field amplitude of 2.4 MV/cm, while the AcOH-MOE film revealed only leaky behavior presumably due to the presence of cracks.

The present results are a good starting point for design of coating solutions containing EG as the alkaline-earth reagent solvent in CSD of more complex BT-based formulations such as lead-free piezoelectric Ba(Zr_x_Ti_1-x_)O_3_-(Ba_1-x_Ca_x_)TiO_3_ films. In these films the control of chemical homogeneity, phase composition and microstructure are the preconditions for the design of their functional properties.

## Data Availability

Not applicable.

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
