# Peer review of "Chemical Solution Deposition of Barium Titanate Thin Films with Ethylene Glycol as Solvent for Barium Acetate"

_molecules, 2022, doi:10.3390/molecules27123753_

Round 1

Reviewer 1 Report

The paper reported chemical solution deposition of BT thin films using EG as solvent for barium acetate, which avoided in-situ water formation and precipitation. Various kind of techniques were used to study the decomposition process and the quality of the as-prepared films. The work is interesting. It can be accepted after addressing the following questions.

1, How to judge intergranular cracks from the SEM images shown in figure 6 and 9. To my opinion, there is no difference between these SEM images. The features assigned by the arrows can still be observed in the SEM images without arrows. In addition, the meaning of the arrows should be pointed out both in the corresponding figure captions and the main text.

2, It is better to given HRTEM images of the films.

3, Why the EG-EtOH film has a higher dielectric permittivity than AcOH-MOE film?

4, For the ferroelectric hysteresis of EG-EtOH film, it is better to give the saturation polarization, remanent polarization, and coercive field in the main text, and compared with the values reported in literatures.

5, In the experimental section, the accelerating voltage of the FE-SEM equipment should be given.

Author Response

Dear Reviewer, thank you for your positive opinion of our manuscript. Please see the attachment of the point-by-point response.

Reviewer 2 Report

There are however a few issues that the authors are kindly requested to take into account for a revised version:

1)    Give the reaction pathway for thin films formation- define mechanisms that are crucial in the understanding of thin film growth.

2)    The thermal decomposition of barium carbonate to produce oxide barium and carbon dioxide. This reaction takes place at a temperature of 850°C. The final temperature of thin films is 800 oC. What is the explanation for this?

3)    Some discussion would be welcome about the formation mechanism of the films.

4)    The substrate for deposition was Pt(111)/TiO2/SiO2/(100)/Si (line 340). Why they are not identified from the XRD diffraction patterns?

5)    Why did you just watch the CO2 and H2O fragments with EGA? because from the decomposition there may be other fragments. All the studies have demonstrated that acetone is the primary volatile product in the decomposition of acetates.  Please add some comments.

6)    Please provide the difference between COO- symmetrical and asymmetrical.

7)    What is the peak-to-valley from AFM imagine?

8)    How was the stability of the solution determined (line 417)?

Finally, the language is in general correct, but some parts require a revision.

Author Response

(The authors gave the same response as above.)

Round 2

Reviewer 2 Report

The answers are correct and complete.

Recommendation: Accept with No Changes